# Learning Consistency-Aware Unsigned Distance Functions Progressively from Raw Point Clouds

**Junsheng Zhou**[1][*]    **Baorui Ma**[1][*]    **Yu-Shen Liu**[1][†]  **Yi Fang**[2]    **Zhizhong Han**[3]

School of Software, BNRist, Tsinghua University, Beijing, China[1]

Center for Artificial Intelligence and Robotics, New York University Abu Dhabi, Abu Dhabi, UAE[2]

Department of Computer Science, Wayne State University, Detroit, USA[3]

`zhoujs21@mails.tsinghua.edu.cn`  `mbr18@mails.tsinghua.edu.cn`

`liuyushen@tsinghua.edu.cn`  `yfang@nyu.edu`  `h312h@wayne.edu`

## Abstract

Surface reconstruction for point clouds is an important task in 3D computer vision. Most of the latest methods resolve this problem by learning signed distance functions (SDF) from point clouds, which are limited to reconstructing shapes or scenes with closed surfaces. Some other methods tried to represent shapes or scenes with open surfaces using unsigned distance functions (UDF) which are learned from large scale ground truth unsigned distances. However, the learned UDF is hard to provide smooth distance fields near the surface due to the non-continuous character of point clouds. In this paper, we propose a novel method to learn consistency-aware unsigned distance functions directly from raw point clouds. We achieve this by learning to move 3D queries to reach the surface with a field consistency constraint, where we also enable to progressively estimate a more accurate surface. Specifically, we train a neural network to gradually infer the relationship between 3D queries and the approximated surface by searching for the moving target of queries in a dynamic way, which results in a consistent field around the surface. Meanwhile, we introduce a polygonization algorithm to extract surfaces directly from the gradient field of the learned UDF. The experimental results in surface reconstruction for synthetic and real scan data show significant improvements over the state-of-the-art under the widely used benchmarks. Project page: `https://junshengzhou.github.io/CAP-UDF`.

## 1 Introduction

Reconstructing surfaces from 3D point clouds is vital in 3D vision, robotics and graphics. It bridges the gap between raw point clouds that can be captured by 3D sensors and the editable surfaces for various downstream applications. Recently, Neural Implicit Functions (NIFs) have achieved promising results by training deep networks to learn Signed Distance Functions (SDFs) [44, 26, 42, 14] or occupancies [40, 46, 41, 10], and then extract a polygon mesh of a continuous iso-surface from a discrete scalar field using the marching cubes algorithm [35]. However, the NIFs approaches based on learning internal and external relations can only reconstruct closed surfaces. The limitation prevents NIFs from representing most real-world objects such as cars with inner structures, clothes with unsealed ends or 3D scenes with open walls and holes.

---

[*]Equal contribution.

[†]The corresponding author is Yu-Shen Liu. This work was supported by National Key R&D Program of China (2022YFC3800600, 2020YFF0304100), and the National Natural Science Foundation of China (62272263, 62072268).

36th Conference on Neural Information Processing Systems (NeurIPS 2022).

As a remedy, state-of-the-art methods [12, 61, 48] learn Unsigned Distance Functions (UDFs) as a more general representation to reconstruct surfaces from point clouds. However, these methods can not learn UDFs with smooth distance fields near surfaces, due to the noncontinuous character of point clouds, even using ground truth distance values or large scale meshes during training. Moreover, most UDF approaches failed to extract surfaces directly from unsigned distance fields. Particularly, they rely on post-processing such as Ball-Pivoting-Algorithm (BPA) [3] to extract surfaces based on the dense point clouds generated from the learned UDF, which is very time-consuming and also leads to surfaces with discontinuity and poor quality.

To solve these issues, we propose a novel method to learn consistency-aware UDFs directly from raw point clouds. We learn to move 3D queries to reach the approximated surface aggressively with a field consistency constraint, and introduce a polygonization algorithm to extract surfaces from the learned unsigned distance functions in a new perspective. Our method can learn UDFs from a single point cloud without requiring ground truth distances, point normals, or a large scale training set. Specifically, given query locations sampled in 3D space as input, We learn to move them to the approximated surface according to the predicted unsigned distances and the gradient at the query locations. More appealing solutions [1, 2, 20, 36] have been proposed to learn SDFs from raw point clouds by optimizing the relationship between the query point and its closest point in raw data as a surface prior. However, since the raw point cloud is a highly discrete approximation of the surface, the closest point to the query location is always inaccurate and ambiguous, which makes the network difficult to converge to an accurate UDF due to the inconsistent or even conflicting optimization directions in the distance field.

Therefore, in order to encourage the network to learn a consistency-aware and accurate unsigned distance field, we propose to dynamically search the optimization target with a specially designed loss function containing field consistency to mimic the conflict optimizations. We also progressively infer the mapping between 3D queries and the approximated zero iso-surface by using well-moved queries as additional priors for promoting further convergence. To extract a surface in a direct way, we propose to use the gradient field of the learned UDFs to determine whether two queries are on the same side of the approximated surface or not. In contrast to NDF [12] which also learns UDFs but takes dense point clouds as output and depends on BPA [3] to generate meshes, our method shows great advantages in efficiency and accuracy due to the straightforward surface extraction.

Our main contributions can be summarized as:

- We propose a novel neural network that learns consistent-aware UDFs directly from raw point clouds. Our method gradually infers the relationship between 3D query locations and the approximated surface with a field consistent loss.

- We introduce an algorithm for directly extracting high-fidelity iso-surfaces with arbitrary topology from the gradient field of the learned unsigned distance functions.

- We obtain state-of-the-art results in surface reconstruction from synthetic and real scan point clouds under the widely used benchmarks.

## 2 Related Works

Surface reconstruction from 3D point clouds has been studied for decades. Classic optimization-based methods [15, 3, 28, 29] tried to resolve this problem by inferring continuous surfaces from the geometry of point clouds. With the rapid development of deep learning [62, 33, 55, 60, 27, 24, 53, 54, 57, 52], the neural networks have shown great potential in reconstructing 3D surfaces [30, 9, 8, 22, 13, 17, 50, 43, 31, 51]. In the following, we will briefly review the studies of deep learning based methods.

### 2.1 Neural Implicit Surface Reconstruction

In the past few years, a lot of advances have been made in 3D surface reconstruction with Neural Implicit Functions (NIFs). The NIFs approaches [7, 25, 46, 40, 16, 26, 32, 41, 34, 44, 39] use either binary occupancies [40, 46, 41, 10] or signed distance functions (SDFs) [44, 26, 42, 14] to represent 3D shapes or scenes, and then use marching cubes [35] algorithm to reconstruct the learned implicit functions into surfaces. Earlier works [40, 44, 25, 10] use an encoder [40, 10] or an optimization

based method [44] to embed the shape into a global latent code, and then use a decoder to reconstruct the shape. To obtain more detailed geometry, some methods [18, 19, 47, 7, 11, 26, 41, 38, 37] proposed to leverage more latent codes to capture local shape priors. To achieve this, the point cloud is first split into different uniform grids [7, 11, 26] or local patches [18, 19, 47], and a neural network is then used to extract a latent code for each grid/patch. Some recent works propose to learn NIFs from a new perspective, such as implicit moving least-squares surfaces [32], differentiable poisson solver [45], iso-points [59] , point convolution [6] or predictive context learning [38]. However, the NIFs approaches can only represent closed shapes due to the characters of occupancies and SDFs.

## 2.2 Learning Unsigned Distance Functions

To model general shapes with open and multi-layer surfaces, NDF [12] learns unsigned distance functions to represent shapes by predicting the unsigned distance from a query location to the continuous surface. However, NDF merely predicts dense point clouds as the output, which requires time-consuming post-processing for mesh generation and also struggles to retain high-quality details of shapes. In contrast, our method is able to extract surfaces directly from the gradient field of the learned UDFs. Following works use image features [61] or query side relations [58] as additional constraints to improve reconstruction accuracy, some other works advance UDFs for normal estimation [48] or semantic segmentation [49]. However, these methods require ground truth distance values or even a large scale meshes during training and are hard to provide smooth distance fields near the surface due to the noncontinuous character of point clouds. While our method does not require any additional supervision but raw point clouds during training, which allows us to reconstruct surfaces for real point cloud scans. In a differential manner, a concurrent work named MeshUDF [21] meshes UDFs from the dynamic gradients during training with a voting schema. On the contrary, we learn a consistancy-aware UDF first and extract the surface from stable gradients during testing. Moreover, our surface extraction algorithm is simpler to use, which is implemented in the marching cube algorithm.

## 2.3 Surface Reconstruction from Raw Point Clouds

Learning implicit functions directly from raw point clouds without ground truth signed/unsigned distance values or occupancy values is more challenge. Current works introduce sign agnostic learning with a specially designed network initialization [1], constraints on gradients [2] or geometric regularization [20] for learning SDFs from raw data. Neural-Pull [36] uses a new way of learning SDFs by pulling nearby space onto the surface. However, they aim to learn signed distances and hence can not reconstruct complex shapes with open or multi-layer surfaces. In contrast, our method is able to learn a continuous unsigned distance function from point clouds, which allows us to reconstruct surfaces for shapes and scenes with arbitrary typology.

# 3 Method

**Problem statement.** We design a neural network to learn UDFs that represents 3D shapes. Given a 3D query location $q = [x, y, z]$, a learned UDF $f$ predicts the unsigned distance value $s = f(q) \in \mathbb{R}$. Current methods depend on ground truth distance values generated from continuous surfaces and employ a neural network to learn $f$ as a regression problem. Different from these methods, we aim to learn $f$ directly from a raw point cloud $P = \{p_i, i \in [1, N]\}$. Furthermore, these methods require post-processing [12] or additional supervision [58] to generate meshes. On the contrary, we introduce an algorithm to extract surfaces directly from $f$ using the gradient field $\nabla f$. The overview of our method is shown in Fig. 1.

## 3.1 Learn UDFs from Raw Point Clouds

We introduce a novel neural network to learn a continuous UDF $f$ from a raw point cloud. We demonstrate our idea using a 2D point cloud $S$ in Fig. 1(a), where $S$ indicates some discrete points of a continuous surface. Specifically, given a set of query locations $Q = \{q_i, i \in [1, M]\}$ which is randomly sampled around $S$, the network moves $q_i$ against the direction of the gradient $g_i$ at $q_i$ with a stride of predicted unsigned distance value $f(q_i)$. The gradient $g_i$ is a vector that presents the partial derivative of $f$ at $q_i = [x_i, y_i, z_i]$, which can be formulated as $g_i = \nabla f(q_i) =$

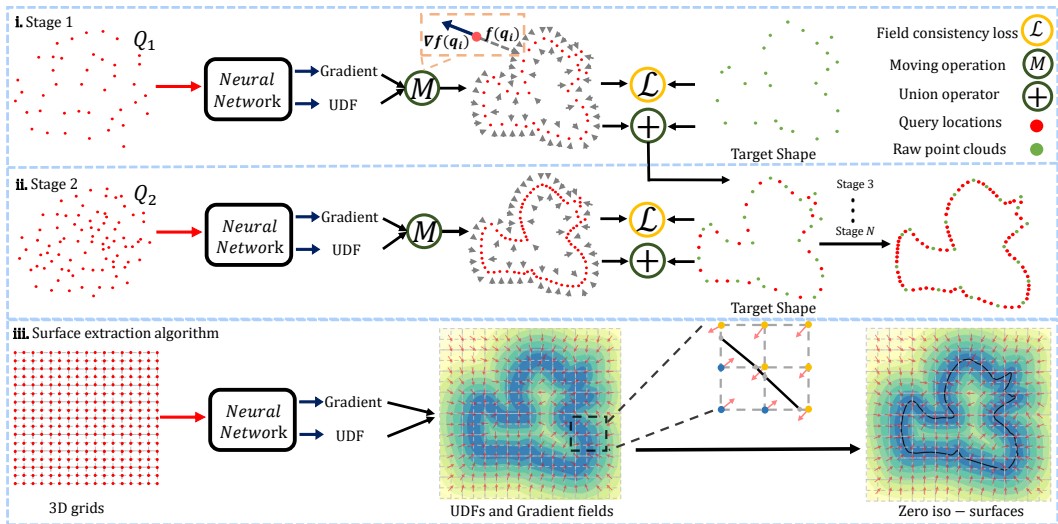

Figure 1: Overview of our method. Given a 3D query $q_i \in Q_1$ as input, the neural network predicts the unsigned distance $f(q_i)$ of $q_i$ and moves $q_i$ against the direction of gradient at $q_i$ with a stride of $f(q_i)$. The field consistency loss is then computed between the moved queries $q'_i$ and the target point cloud $P$ as the optimization target. After the network converges in the current stage, we update $P$ with a subset of $q'_i$ as additional priors to learn more local details in the next stage. Finally, we use the gradient field of the learned UDFs to model the relationship between different 3D grids and extract iso-surfaces directly.

$[\partial f / \partial x, \partial f / \partial y, \partial f / \partial z]$. The direction of $g_i$ indicates the orientation of where the unsigned distance increases the fastest in 3D space, which points the direction away from the surface, therefore moving $q_i$ against the direction of $g_i$ will find a path to reach the surface of $S$. The moving operation can be formulated as:

$$z_i = q_i - f(q_i) \times \nabla f(q_i)/||\nabla f(q_i)||_2, \tag{1}$$

where $z_i$ is the location of the moved query $q_i$, and $\nabla f(q_i)/||\nabla f(q_i)||_2$ is the normalized gradient $g_i$, which indicates the direction of $g_i$. The moving operation is differentiable in both the unsigned distance value and the gradient, which allows us to optimize them simultaneously during training.

The four examples in Fig. 2 show the distance fields learned by Neural-Pull [36], SAL [1], NDF [12] and our method for a sparse 2D point cloud $P$ which only contains 13 points. One main branch to learn signed or unsigned distance functions for point clouds is to directly minimize the mean squared error between the predicted distance value $f(q_i)$ and the euclidean distance between $q_i$ and its nearest neighbour in $P$, as proposed in NDF and SAL. However, as shown in Fig. 2(c), NDF leads to an extremely discrete distance field. To learn a continuous distance field, NDF introduces ground truth distance values extracted from the continuous surface as extra supervision, which prevents it from learning directly from raw point clouds. SAL shows a great capacity in learning SDFs for watertight shapes using a carefully designed initialization. However, as shown in Fig. 2(b), SAL fails to converge to a multi-structure shape since the network is initialized as a single layer shape prior. Neural-Pull uses a similar way as ours to pull queries onto the surface, thus also learns a continuous signed distance field as shown in Fig. 2(a). However, the nature of SDF prevents Neural-Pull from reconstructing open surfaces like the "1" on the left of Fig. 2(a). As shown in Fig. 2(d), our method can learn a continuous level set of distance field and can also represent open surfaces.

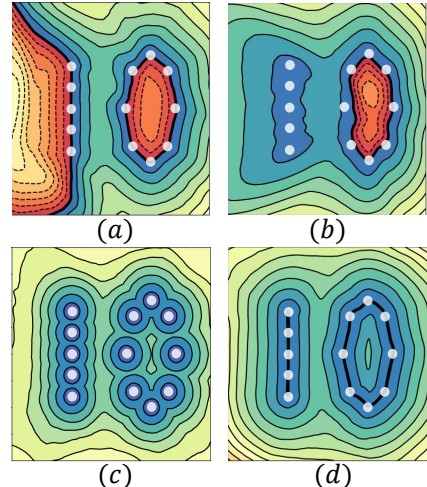

Figure 2: The level-sets show the distance fields learned by (a) Neural-Pull, (b) SAL, (c) NDF, (d) Ours. The color of blue and red represent positive or negative distance. The darker the color, the closer it is to the approximated surface.

One way to extend Neural-Pull directly to learn UDFs is to predict a positive distance value for each query and pull it to the nearest neighbour in $P$. However, for shapes with complex topology, this optimization is often ambiguous due to the noncontinuous character of raw point clouds. We solve this problem by introducing consistency-aware field learning.

## 3.2 Consistency-Aware Field Learning

Neural-Pull leverages a mean squared error to minimize the distance between the moved query $z_i$ and the nearest neighbour $n_i$ of $q_i$ in $P$:

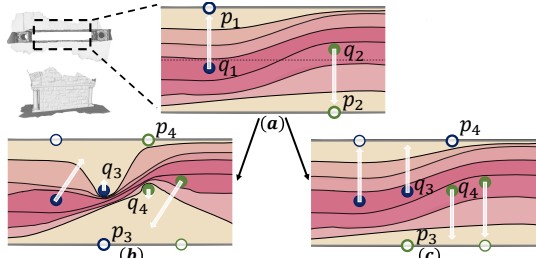

$$\mathcal{L} = \frac{1}{M} \sum_{i \in [1,M]} ||z_i - n_i||_2^2. \quad (2)$$

However, the direct optimization of the loss in Eq. (2) will form a distorted field and lead some queries to get stuck due to the conflict optimization which makes the network difficult to con-

Figure 3: Demonstration case of different losses.

verge. We show a 2D case of learning UDFs for a double-deck wall using the loss Eq. (2) as in Fig. 3(b) and using our loss as in Fig. 3(c). Assume $p_1$ and $p_2$ as two discrete points in two different decks of the wall, $q_1$ and $q_2$ are two queries whose closest neighbours are $p_1$ and $p_2$, respectively. Optimizing the network using $q_1$ and $q_2$ by minimizing Eq. (2) or our proposed loss in Eq. (3) will lead to an unsigned distance field as in Fig. 3(a). Assuming in the next training batch, $q_3$ and $q_4$ are two queries whose closest neighbours are $p_3$ and $p_4$. If we use the loss in Eq. (2), the optimization target of $q_3$ is to minimize $\mathcal{L} = ||z_3 - p_3||_2^2$. Notice that the target point $p_3$ is located on the lower surface, however the opposite direction of gradient around $q_3$ is upward at this moment. Therefore, the partial derivative $\frac{\partial \mathcal{L}}{\partial z_3}$ leads to a decrease in the unsigned distance value $f(q_3)$ predicted by the network. The case of $q_4$ is optimized similarly. An immediate consequence is that the inconsistent optimization directions will form a distorted fields that has local minima of unsigned distance values at $q_3$ and $q_4$ as in Fig. 3(b). However, this situation causes other query points around point $q_3$ or $q_4$ to get stuck in the distorted fields and unable to move to the correct location, thus making the network hard to converge.

To address this issue, we propose a loss function which can keep the consistency of unsigned distance fields to avoid the conflicting optimization directions. Specifically, instead of strictly constraining the convergence target before forward propagation as Eq. (2), we first predict the moving path of a query location $q_i$ and move it using Eq. (1) to $z_i$, then look for the surface point $p_i$ in $P$ which is the closest to $z_i$ and minimize the distance between $z_i$ and $p_i$. As shown in Fig. 3(c), after moving $q_3$ against the gradient direction with a stride of $f(q_3)$ to $z_3$, the closest surface point of $z_3$ lies on the upper deck, so the distance fields remain continuous and are optimized correctly. In practical, we use chamfer distance as a suitable loss implementation, formulated as:

$$\mathcal{L}_{\mathrm{CD}} = \frac{1}{M} \sum_{i \in [1,M]} \min_{j \in [1,N]} ||z_i - p_j||_2 + \frac{1}{N} \sum_{j \in [1,N]} \min_{i \in [1,M]} ||p_j - z_i||_2. \quad (3)$$

We also use toy examples as shown in Fig. 4 to show the advantage of our proposed field consistency loss. We learn UDFs for a raw point cloud of a double-deck wall as shown in Fig. 4(a). Fig. 4(b) denotes the randomly sampled query locations between two decks of the wall where the different colors mean the queries are closer to the upper or lower deck of the wall. Fig. 4(c) and Fig. 4(d) indicate the moved queries by loss in Eq. (2) and Eq. (3). It can be seen that our proposed loss can move most of the queries to the correct surface position, and Neural-Pull loss stops moving in many places or moves queries to the wrong places due to the field inconsistency in optimization. Fig. 4(e) and Fig. 4(f) show the learned distance field of a car with inner structure by the loss in Eq. (2) and our loss.

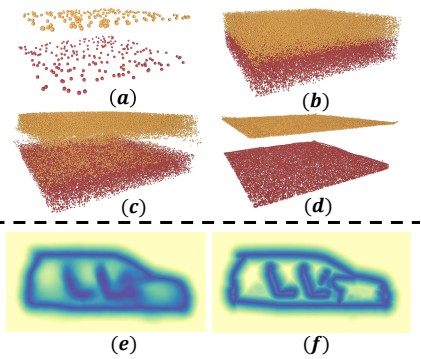

Figure 4: Advantages of our loss.

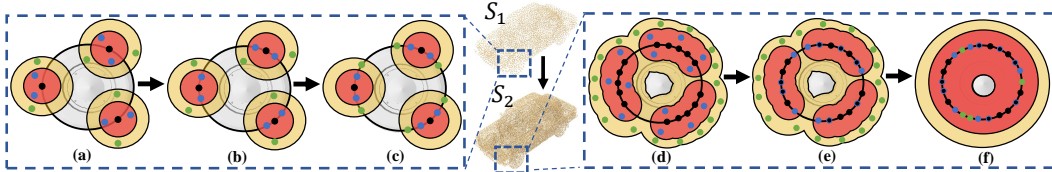

Figure 5: Illustration of progressively approximating the surface.

## 3.3 Progressive Surface Approximation

Moreover, in order to predict unsigned distance values more accurately and learn more local details, we propose a progressive learning strategy by taking the intermediate results of moved queries as additional priors. Given a raw point cloud which is a discrete representation of the surface, we have made a reasonable assumption: the closer the query location is to the given point cloud, the smaller the error of searching the target point on the given point cloud. We provide the Proof of this assumption in the appendix. Based on this assumption, we set up two regions: the high confidence region with small error and the low confidence region with large error. We sample query points in the high confidence region to help train the network and sample auxiliary points in the low confidence region to move to the estimated surface position by network gradient after network convergence at current stage, where the moved auxiliary points are regarded as the surface prior for the next stage. Notably, the auxiliary points do not participate in network training since these points with low confidence will lead to a large error and affect network training. Since the low confidence regions which are not optimized explicitly during training are distributed interspersed between the high confidence regions, according to the integral Monotone Convergence Theorem [5], the UDFs and gradient predicted by the low confidence region are a smooth expression of the trained high confidence region. We save the moved queries and auxiliary points and use them to update $S$. According to the updated point cloud, we re-divide the regions with high confidence and low confidence and re-sample the query points and auxiliary points for the next stage.

We demonstrate our idea using a 2D case in Fig. 5.(a). We divided the regions with high confidence (red region) and low confidence (yellow region) based on the given raw point cloud $S_1$ (black dots) and then sample query points $Q_1 = \{q_i, i \in [1, M]\}$ (blue dots) and auxiliary points(green dots) $A_1 = \{a_i, i \in [1, M]\}$. (b) We train the network to learn UDFs by moving the query locations $Q_1$ using Eq. (1), and optimize the network by minimizing Eq. (3). (c) After network convergence at current stage, we move query points $Q$ and auxiliary points $A$ to the estimated surface position by the gradient of network, $S_1' = p - f(p) \times \nabla f(p)/||\nabla f(p)||_2, p \subset Q_1 \cup A_1$. (d) We save the moved points $S_1'$ and use them to update $S$, $S_2 = S_1 \cup S_1'$. According to the updated point cloud $S_2$, we re-divided the regions with high confidence and low confidence and re-sampled the query points $Q_2$ and auxiliary points $A_2$. (e) We continue to train the network by moving query points $Q_2$ to the updated $S_2$, and then update $S$ by combining the moved $Q_2$ and $A_2$ with $S_2$. (f) Because of the more continuous surface prior information, the network will learn more accurately and learn more local details of the UDFs.

## 3.4 Surface Extraction Algorithm

Unlike SDFs, UDFs fail to extract surfaces by marching cubes since UDFs cannot perform inside/outside tests on 3D grids. To address this issue, we propose to use the gradient field $\nabla f$ to determine whether two 3D grid locations are on the same side or the opposite side of the surface approximated by the point clouds $P$. We make an assumption that on a micro-scale of the surface, the space can always be divided into two sides, where the 3D query locations of different sides denoted as $Q_{in} = \{q_{in}^i, i \in [1, L]\}$ and $Q_{out} = \{q_{out}^i, i \in [1, I]\}$. For two queries $q_{in}^i$ and $q_{out}^j$ in different sides of the surface, the included angle between

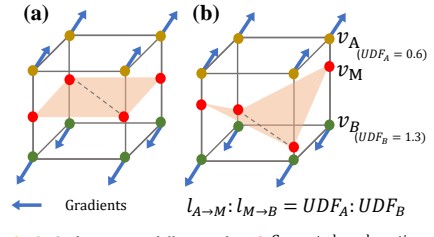

Figure 6: Surface extraction algorithm.

the directions of the gradients $\nabla f(q_{in}^i)$ and $\nabla f(q_{out}^j)$ are always more than 90 degrees, which can be formulated as $\nabla f(q_{in}^i) \cdot \nabla f(q_{out}^j) < 0$. On the contrary, for two queries $q_{in}^i$ and $q_{in}^j$ in the same side, the formula $\nabla f(q_{in}^i) \cdot \nabla f(q_{in}^j) > 0$ holds true. So, we can classify whether two points

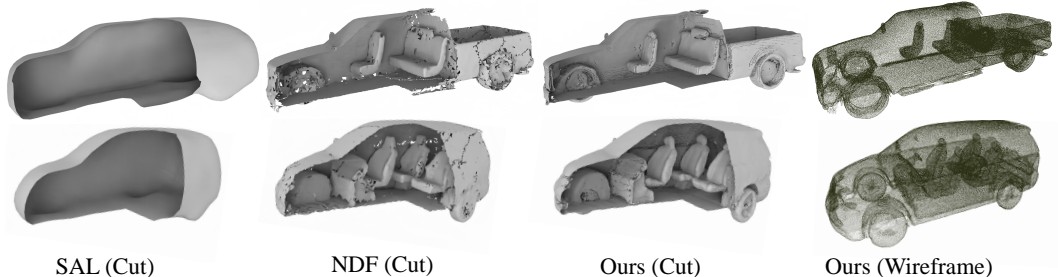

| SAL (Cut) | NDF (Cut) | Ours (Cut) | Ours (Wireframe) |

Figure 7: Visual comparisons of surface reconstruction on ShapeNet cars.

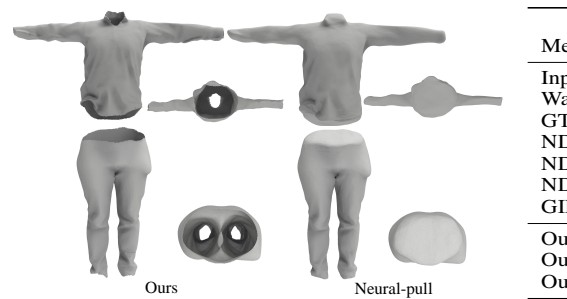

| Ours | Neural-pull |

Figure 8: Visual comparisons on MGD dataset.

Table 1: Comparison on ShapeNet cars.

| Method | Chamfer-L2 | | F-Score | |
| | Mean | Median | $F1^{0.005}$ | $F1^{0.01}$ |
|---|---|---|---|---|
| Input | 0.363 | 0.355 | 48.50 | 88.34 |
| Watertight GT | 2.628 | 2.293 | 68.82 | 81.60 |
| GT | 0.076 | 0.074 | 95.70 | 99.99 |
| $\text{NDF}_{BPA}$[12] | 0.202 | 0.193 | 77.40 | 97.97 |
| $\text{NDF}_{gradRA}$ | 0.160 | 0.152 | 82.87 | 99.35 |
| $\text{NDF}_{PC}$ | 0.126 | 0.120 | 88.09 | 99.54 |
| GIFS [58] | 0.128 | 0.123 | 88.05 | 99.31 |
| $\text{Ours}_{BPA}$ | 0.141 | 0.138 | 84.84 | 99.33 |
| $\text{Ours}_{gradRA}$ | **0.119** | **0.114** | **88.55** | **99.82** |
| $\text{Ours}_{PC}$ | **0.110** | **0.106** | **90.06** | **99.87** |

are in the same or the opposite side using dot product of gradients, $cls(q_i, q_j) = \nabla f(q_i) \cdot \nabla f(q_j)$. Based on that, we divide the space into 3D grids (e.g. $256^3$), and perform gradient discrimination on the 8 vertices $v_i$ in each cell grid according to the gradient field $\nabla f$ using $cls(q_i, q_j)$. As shown in Fig. 6(a), the gradient field separates the vertices into two sets, where we can further adapt marching cubes algorithm [35] to create triangles for the grid using the lookup table. The complete surface is generated by grouping triangles of each grid together. To accelerate the surface extraction process and avoid extracting unexpected triangles in the multi-layers structures, we set a threshold $\theta$ to stop surface extraction on grids where $f(g_i) > \theta, i \in [0, 7]$.

**Mesh refinement.** The initial surface extracted by marching cubes is only a discrete approximation of the zero iso-surface. To achieve a more detailed mesh, we propose to refine it using the UDF values. As shown in Fig. 6(b), given the predicted UDF values $UDF_A$ and $UDF_B$ of grid vertex $v_A$ and $v_b$, the mesh vertex $v_M$ can be moved to a finer position where $l_{A \to M} : l_{M \to B} = UDF_A : UDF_B$.

# 4 Experiments

We evaluate our method on the task of surface reconstruction from raw point clouds. We first demonstrate the ability of our method to reconstruct general shapes with open and multi-layer surfaces in Sec.4.1. Next, we apply our method to reconstruct surfaces for real scanned raw data including 3D objects in Sec.4.2 and complex scenes in Sec.4.3. Ablation studies are shown in Sec.4.4.

**Implementation details.** To learn UDFs for raw point clouds $P$, we adopt a neural network similar to OccNet [40] to predict the unsigned distance given 3D queries as input. Our network contains 8 layers of MLP where each layer has 256 nodes. Similar to Neural-Pull and SAL, given the single point cloud $P$ as input, we do not leverage any condition and overfit the network to approximate the surface of $P$ by minimizing the loss of Eq. (3). Therefore, we do not need to train our network on large scale training dataset in contrast to previous methods [12, 58, 26]. In addition, we use the same strategy as Neural-Pull to sample 60 queries around each point $p_i$ on $P$ as training data. A Gaussian function $\mathcal{N}(\mu, \sigma^2)$ is adopt to calculate the sampling probability where $\mu = p_i$ and $\sigma$ is the distance between $p_i$ and its 50-th nearest points on $P$. For sampling auxiliary points in the low confidence region, the standard deviation is set to $1.1\sigma$. And we train our network for two stages in practice.

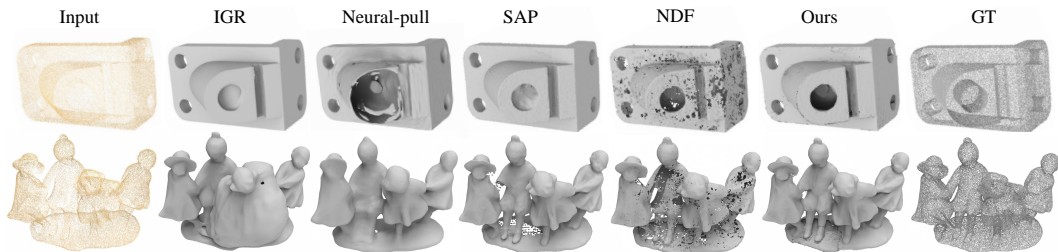

| Input | IGR | Neural-pull | SAP | NDF | Ours | GT |

Figure 9: Visual comparisons of surface reconstruction on the SRB dataset.

## 4.1 Surface Reconstruction for Synthetic Shapes

**Dataset and metrics.** For the experiments on synthetic shapes, we follow NDF [12] to choose the "Car" category of the ShapeNet dataset which contains the greatest amount of multi-layer shapes and non-closed shapes. And 10k points is sampled from the surface of each shape as the input. Besides, we employ the MGD dataset [4] to show the advantage of our method in open surfaces. To measure the reconstruction quality, we follow GIFS [58] to sample 100k points from the reconstructed surfaces and adopt the Chamfer distance ($\times 10^4$), Normal Consistency (NC) [40] and F-Score with a threshold of 0.005/0.01 as evaluation metrics.

**Comparison.** We compare our method with the state-of-the-art works NDF [12] and GIFS [58]. We quantitatively evaluate our method with NDF and GIFS in Tab. 1. We also report the results of points sampled from the watertight ground truth (watertight GT in table) as the upper bound of the traditional SDF-based or Occupancy-based implicit

| Method | Chamfer-L2 | F-Score$^{0.01}$ | NC |
|---|---|---|---|
| Neural-Pull [36] | 4.447 | 94.49 | 91.83 |
| NDF [12] | 0.658 | 76.11 | 92.84 |
| Ours | 0.117 | 99.68 | 97.80 |

Table 2: Comparisons on MGD dataset.

functions. To show the superior limit of this dataset, we sample two different sets of points from the ground truth mesh and report their results (GT in table). For a comprehensive comparison with NDF, we transfer our gradient-based reconstruction algorithm to extract surfaces from the learned distance field of NDF, and report three metrics of NDF and our method including generated point cloud ($*_{PC}$), mesh generated using BPA ($*_{BPA}$) and mesh generated using our gradient-based reconstruction algorithm ($*_{gradRA}$). As shown in Tab. 1, we achieve the best results in terms of all the metrics. Moreover, our gradient-based reconstruction algorithm shows great generality in transferring to the learned gradient field of other method (e.g. NDF) by achieving significant improvement over the traditional method (BPA). We also provide the results of surface reconstruction on MGD [4] dataset as shown in Tab. 2, where we largely outperform other methods.

We further present a visual comparison with SAL and NDF in Fig. 7. Previous methods (e.g. SAL) take SDF as output and are therefore limited to single-layer shapes where the inner-structure is lost. NDF learns UDFs and is able to represent general shapes, but it outputs a dense point cloud and requires BPA to generate meshes, which leads to an uneven surface. On the contrary, we can extract surfaces directly from the learned UDFs, which are continuous surfaces with high fidelity. We also provide a visual comparison with Neural-Pull in MGD dataset as shown in Fig. 8, where we accurately reconstruct the open surfaces but Neural-Pull fails to keep the original geometry.

## 4.2 Surface Reconstruction for Real Scans

**Dataset and metrics.** For surface reconstruction of real point cloud scans, we follow SAP to evaluate our methods under the Surface Reconstruction Benchmarks (SRB) [56]. We use Chamfer distance and F-Score with a threshold of 1% for evaluation. Note that the ground truth is dense point clouds.

**Comparison.** We compare our method with state-of-the-art classic and data-driven surface reconstruction methods in the real scanned SRB dataset, including IGR [20], Point2Mesh [23], Screened Poisson Surface Reconstruction (SPSR) [29], Shape As Points (SAP) [45], Neural-Pull [36] and NDF [12]. The numerical comparison is shown in Tab. 4, where we achieve the best accuracy. The visual comparisons in Fig. 9 demonstrate that our method is able to reconstruct a continuous surface with local geometry consistence while other methods struggle to reveal the geometry details. For example, IGR, Neural-Pull and SAP mistakenly mended or failed to reconstruct the hole of the anchor while our method is able to keep the correct geometry.

| Method | 100/$m^2$ | | 500/$m^2$ | | 1000/$m^2$ | |
|---|---|---|---|---|---|---|
| | L2CD | L1CD | L2CD | L1CD | L2CD | L1CD |
| ConvONet [46] | 7.859 | 0.043 | 13.192 | 0.052 | 14.097 | 0.052 |
| LIG [20] | 6.265 | 0.049 | 5.633 | 0.048 | 6.190 | 0.048 |
| DeepLS [7] | 3.029 | 0.044 | 6.794 | 0.050 | 1.607 | 0.025 |
| $NDF_{PC}$ [12] | 0.409 | 0.012 | 0.377 | 0.014 | 0.561 | 0.017 |
| $NDF_{mesh}$ | 0.452 | 0.014 | 0.475 | 0.016 | 0.872 | 0.022 |
| OnSurf [37] | 1.154 | 0.021 | 0.862 | 0.020 | 0.706 | 0.020 |
| $Ours_{PC}$ | **0.144** | **0.010** | **0.078** | **0.009** | **0.072** | **0.010** |
| $Ours_{mesh}$ | **0.187** | **0.011** | **0.122** | **0.010** | **0.121** | **0.009** |

Table 3: Surface Reconstruction under 3D Scene, L2CD×1000.

| Method | Chamfer-L1 | F-Score |
|---|---|---|
| IGR [20] | 0.178 | 75.5 |
| Point2Mesh [23] | 0.116 | 64.8 |
| SPSR [29] | 0.232 | 73.5 |
| SAP [45] | 0.076 | 83.0 |
| Neural-Pull [36] | 0.106 | 79.7 |
| $NDF_{PC}$ [12] | 0.185 | 72.2 |
| $NDF_{mesh}$ | 0.238 | 68.6 |
| $Ours_{PC}$ | **0.068** | **90.4** |
| $Ours_{mesh}$ | **0.073** | **84.5** |

Table 4: Comparisons on SRB.

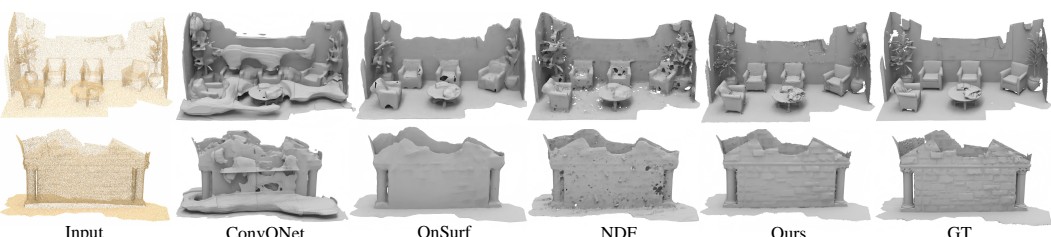

| Input | ConvONet | OnSurf | NDF | Ours | GT |

Figure 10: Visual comparison with different methods under 3D Scene. Inputs contains 1k points/$m^2$.

## 4.3 Surface Reconstruction for Scenes

**Dataset and metrics.** To further demonstrate the advantage of our method in surface reconstruction of real scene scans, we follow OnSurf [37] to conduct experiments under the 3D Scene dataset [63]. Note that the 3D Scene dataset is a challenging real-world dataset with complex topology and noisy open surfaces. We uniformly sample 100, 500 and 1000 points per $m^2$ at the original scale of scenes as the input and follow OnSurf to sample 1M points on both the reconstructed and the ground truth surfaces. We leverage L1 and L2 Chamfer distance to evaluate the reconstruction quality.

**Comparison.** We compare our method with the state-of-the-arts scene reconstruction methods ConvONet [46], LIG [26], DeepLS [7], OnSurf [37] and NDF [12]. The numerical comparisons in Tab. 3 show that our method significantly outperform the other methods under different point densities. The visual comparisons in Fig. 10 further shows that our reconstructions present more geometry details in complex real scene scans. Note that all the other methods have been trained in a large scale dataset, from which they gain additional prior information. On the contrary, our method does not leverage any additional priors or large scale training datasets, and learns to reconstruct surfaces directly from the raw point cloud, but still yields a non-trivial performance.

## 4.4 Ablation Study

We conduct ablation studies to justify the effectiveness of each design in our method and the effect of some important parameters. We report the performance in terms of L2-CD under a subset of the ShapeNet Car dataset. By default, all the experimental settings are kept the same as in Sec. 4.1, except for modified part described in each ablation experiment below.

**Framework design.** We first justify the effectiveness of each design of our framework in Tab. 5. We first directly use the loss proposed in Neural-Pull and find that the performance degenerates dramatically as shown by "NP loss".

| $\times 10^4$ | NP loss | Exponent | Scratch | Ours |
|---|---|---|---|---|
| L2CD | 0.2381 | 0.1218 | 0.1497 | **0.1112** |

Table 5: Effect of framework design.

We also use $g(x) = 1 - e^{(-x)}$ to replace $g(x) = |x|$ on the last layer of the network for $f$ before output, but found no improvement as shown by "Exponent". We train the second stage from scratch as shown by "Scratch" and prove that an end-to-end training strategy is more effective.

**The effect of stage numbers.** The number of stages during progressive surface approximation is also a crucial factor in the network training. We report the performance of training our network in different number of stages $St = [1, 2, 3, 4]$ in Tab. 6. We start the training of next stage after the previous one converges.

| $\times 10^4$ | 1 | 2 | 3 | 4 |
|---|---|---|---|---|
| L2CD | 0.1218 | **0.1112** | 0.1107 | 0.1107 |

Table 6: Effect of stage numbers.

We found that two stages training brings great improvement than training a single stage, and the improvements with 3rd and 4th stages are subtle.

**The effect of low confidence range.** We further explore the range of confidence region sample. Assume $\sigma$ as the range of high confidence region, we use $0.9\sigma$, $1.0\sigma$, $1.1\sigma$ and $1.2\sigma$ as the range of low confidence

| $\times 10^4$ | 0.9 | 1.0 | 1.1 | 1.2 |
|---|---|---|---|---|
| L2CD | 0.1133 | 0.1130 | **0.1112** | 0.1131 |

Table 7: Effect of low confidence range.

region. The results in Tab. 7 show that a too small or too large range will degenerate the performance.

**Surface extraction.** We evaluate the effect of mesh refinement and the performance of different 3D grid resolutions. Tab. 8 shows the accuracy and efficiency of different resolutions. We observe that the mesh refinement highly improves the accuracy and higher resolutions leads to better reconstructions at a cost of speed.

| $\times 10^4$ | $64^3$ | $128^3$ | $256^3$ | $320^3$ |
|---|---|---|---|---|
| w/o refine | 0.4169 | 0.1738 | 0.1294 | 0.1238 |
| refine | 0.1606 | 0.1174 | **0.1112** | 0.1105 |
| Time | 3.0 s | 21.9 s | 162.2 s | 307.6 s |

Table 8: Ablations on surface extraction.

## 5 Conclusion

We propose a novel method to learn continuous UDFs directly from raw point clouds by learning to move 3D queries to reach the approximated surface progressively. Our introduced reconstruction algorithm can extract surfaces directly from the gradient fields of the learned UDFs. Our method does not require ground truth distance values or point normals, and can reconstruct surfaces with arbitrary topology. One limitation of our method is that we use uniformly divided grids to extract surface, which can be improved with a coarse-to-fine paradigm.

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
