# OpenReview forum: "Learning Consistency-Aware Unsigned Distance Functions Progressively from Raw Point Clouds"
_NeurIPS.cc/2022/Conference — NeurIPS 2022 Accept_

### Official Review · Reviewer_Aixv · 2022-07-10

**Rating:** 7
**Confidence:** 3
**Soundness:** 3 good
**Presentation:** 3 good
**Contribution:** 3 good

**Summary:**

This paper presents a framework to learn Unsigned Distance Functions (UDF) from point clouds. The learned continuous UDF can then be used to extract surface to represent 3D geometry. One of the challenges of learning a continuous UDF from a discrete point cloud is the instability of gradient due to the sparsity of points. To this end, the authors propose a novel loss function with a field consistency constrain. They also designed a progressive scheme to learn more local details. Unlike SDF that can recover surfaces using the Marching Cubes algorithm directly, UDF cannot pass the inside-outside test due to the lack of direction information (i.e., sign). Therefore, this paper propose to use the relative angle between query points to test whether they cross the iso-surface. Experiments demonstrate the proposed method outperform existing methods and ablation studies verify the design choices.

**Questions:**

N/A

**Strengths And Weaknesses:**

Strengths:
- The paper is well written. It is easy to follow

- The figures are greatly helpful for readers to understand the idea.

- The proposed idea is interesting and effective as it is supported by the superior performance in comaprisons against existing methods. Furthermore, ablation studies are sufficient to validate design choices.

Weaknesses:

- figure captions
I would recommend to expand figure captions so that readers don't need to jump back and forth between text and figure.

---

> ### Author Response · Authors · 2022-08-02
> **Response to Review Aixv**
>
> We thank the reviewer for considering our method interesting, promising and novel.  And we are pleased to hear the figures helpful and ablation studies convincing. We will expand figure captions with detailed descriptions in revision. Thanks for your suggestions!

---

> > ### Author Response · Authors · 2022-08-08
> > **Looking Forward to Hearing from Reviewer Aixv**
> >
> > Dear Reviewer Aixv,
> >
> > Following your questions, we will expand figure captions with detailed descriptions in revision. We would like to know whether you believe we have addressed your concerns, and please let us know if you have any other questions.
> >
> > Thanks for your time,
> >
> > The Authors

---

### Official Review · Reviewer_tXas · 2022-07-12

**Rating:** 8
**Confidence:** 4
**Soundness:** 3 good
**Presentation:** 3 good
**Contribution:** 3 good

**Summary:**

This paper proposed a method for surface reconstructions by training a neural network to predict unsigned distance fields (UDF). The learned UDFs are consistent-aware, and can be trained without ground truth distance fields, point normals, or large scale training datasets. A high quality surface can be extracted from the gradient vector field of the learned UDFs. The paper has achieved appealing results compared to some of the state of the art algorithms.

**Questions:**

1) Are there plans to release source code or pretrained models to the community?
2) The paper did not talk much about the scalability of the proposed method. For example, it would be interesting to know if the proposed method can handle millions of points, city-scale LiDAR scans, etc? How much computation time/computation resources does the proposed method need?

**Limitations:**

The author addressed the limitations of uniformly dividing grids for surface extraction.

**Strengths And Weaknesses:**

1) The paper carefully examined the current failure mode of the UDF approximation methods, thus proposing the consistency-aware field learning loss, and the progressive approximation paradigm. These strategies greatly improved the quality of the learned UDF, as illustrated by the paper.
2) Traditional marching cube algorithms cannot be directly applied on UDFs since there is no inside/outside information in an UDF. The paper proposed a novel surface extraction algorithm by looking at the gradient vector field of the learned UDF. From the originality and quality perspective, the paper has done well.
3) Presentation is well done, language and visualization are clear.
4) From a significance perspective, the reviewer believe the paper has boosted the SOTA by a quite large margin. The reconstructed surface has much higher quality in many challenging scenarios.

---

> ### Author Response · Authors · 2022-08-02
> **Response to Review tXas**
>
> We appreciate that the reviewer finds our paper promising, novel, and well-written. We address additional comments below.
>
> **Q1: Are there plans to release source code or pretrained models to the community?**
>
> Yes. We will make source code and pretrained models public within 2 weeks after acceptance.
>
> **Q2: The paper did not talk much about the scalability of the proposed method.**
>
> As suggested, we will add more discussions of the scalability of our method in the conclusion.
>
> **Q3: Can the proposed method handle millions of points, city-scale LiDAR scans, etc?**
>
> We believe the answer is yes if we adopt the sliding window strategy to reconstruct surfaces part by part. Due to the catastrophic forgetting problem of the neural networks, it is extremely difficult to represent large-scale scenes within a single network. To solve this issue, recent works (e.g. DeepLS [ECCV 2020] and BlockNeRF [CVPR 2022]) propose to use the sliding window strategy to represent large scale scenes using separate parts and have shown promising results. We also consider this as an interesting future work to transfer the sliding window strategy to our method for representing large scale data and thanks for pointing it out! We will add it to the future work.
>
> **Q4: How much computation time/computation resources does the proposed method need?**
>
> Thanks for the question. We make a comparison with Neural-Pull, IGR, Point2mesh on the computational cost of optimizing for a single point cloud in the following table:
>
> |methods|Neural-Pull|IGR|Point2mesh|Ours|
> |:-:|:-:|:-:|:-:|:-:|
> |Time (s)|1150|1212|4028|**667**|
> |Memory (GB)|2.2|6.1|5.2|**2.0**|
>
> The optimization time is evaluated on a single GTX 3090 GPU. It shows that our method converges faster than all the baselines. We will include the table in the supplementary, and we also provided the efficiency comparison of surface generation in Table 2 of the supplementary.

---

> > ### Author Response · Authors · 2022-08-08
> > **Looking Forward to Hearing from Reviewer tXas**
> >
> > Dear Reviewer tXas,
> >
> > Following your questions, we provided additional explanations on scaling our method to large-scale scenes and reported the computational cost compared to the state-of-the-arts to illustrate the scalability and efficiency of our approach. And we are willing to release source code and pretrained models within 2 weeks after acceptance.
> >
> > We would like to know whether you believe we have addressed your concerns, and please let us know if you have any other questions.
> >
> > Thanks for your time,
> >
> > The Authors

---

### Official Review · Reviewer_1bdf · 2022-07-12

**Rating:** 6
**Confidence:** 4
**Soundness:** 3 good
**Presentation:** 1 poor
**Contribution:** 3 good

**Summary:**

This paper presents a method to mesh point clouds. It performs optimization on a single scene (without any training) . It claims 2 contributions:
1. a loss function and optimization strategy, which in my understanding is essentially the one presented in neural pull [26] for signed distance function used for unsigned distance function and symetrized. It is often refered to as a "consistency aware" / "field consistency loss" and as fighting "adversarial optimization" (which makes little sense to me)
2. a meshing strategy, which to me seems an adaptation of marching cube to unsigned distance function
I would say there is a 3rd contribution, which is not claimed in the intro but is a part of the method section, which is the progressive (i.e. 2 step in practice) surface approximation, even if the quantitative gains associated to it are small.
It presents results on several dataset that seem to improve state of the art

**Questions:**

Please address in details may questions in weaknesses 1 and 2, and explain how you will modify the paper to clarify it (if I am unconvinced with the answer on these points I am likely to change my rating)

**Limitations:**

yes

**Strengths And Weaknesses:**

While I am not an expert of the area, the benefits of the proposed approach in term of results seem clear to me, which I think is the main strength of the paper. The proposed approach also seem to make a lot of sense and is quite simple.

I see several weaknesses in the paper:
1. I found the paper very hard to parse while the proposed approach is quite simple:
- this is particularly true for 3.1 and 3.2. I think this is written only for people who are very familiar with the 3 most related papers + has far too many forward and backward pointers. For example, in 3.1, before anything about the method has been explained (no loss function, nothing on optimization), there are results, comparisons with 3 baselines and discussion of the differences (l 126-151 and figure 2) I do not think it can make sense before the full paper has been read and understood. Similarly, l. 167 discusses results obtained when using equation 3 which is presented l. 185. If this was a journal submission this could easily be solved with a "major revision", for a conference paper this is much harder to trust the authors with a major rewriting of the paper...
- another thing that annoyed me is that I could understand none of what the paper was doing from the abstract and intro. Terms like "consistency aware" / "field consistency loss" and as fighting "adversarial optimization" are not explained while  they refer to very simple idea, and I think they are designed to impress but make little sense/are not adapted (not sure if it's the fault of the authors or if they re-use terms from other paper)

2. I am unsure what the real technical contribution are:
- to me the first contribution, which is a big part of the method section (3.1 and 3.2), is actually a very small modification of neural pull [26]. I think this is not recognized enough in the paper and find that a very annoying issue.
- the progressive surface approximation seem novel but this is not claimed clearly, so I am unsure whether this might be following another paper
- the surface extraction seem to be a relatively simple adaptation of marching cube: if the authors agree, this again should be acknowledge much more clearly in the abstract, intro and 3.4
- to me, the real contribution is actually taking the previous small idea together and making them into a very effective algorithm, which could make for a great paper if only it was acknowledged better and each part explained much more simply. Unfortunately, this again put me on the verge of recommending rejection for a conference paper.

3. smaller concerns are associated with the experiments (which again I found in general convincing)
- since the output is a mesh I would like to see metrics related to meshes, not only point cloud. For example, it would be quite easy to measure normal distances (up to flip)
- I am confused by the "low confidence range" experiment (table 7) I guess the low confidence range should be understood as "in addition to sigma", so 0.9\sigma for  example actually means "between 1 and 1.9\sigma from the origin": is that right? If that's right, why not experiment with much smaller values (e.g. 0.1 and 0.5 sigma), and in any case with larger values (2 and 4 sigma)? That would make the trend much clearer. In any case, this should be better explained, a small figure (earlier, in the method section) could help

All in all, because I think the method makes sense and because (as far as I can judge not being an expert) the results seem very good and the ablation convincing, I would still tend to recommend accepting the paper, trusting the authors with a major rewriting.

---

> ### Author Response · Authors · 2022-08-02
> **Response to Review 1bdf**
>
> We thank the reviewer for his constructive feedback. In the following, we address the main concerns and explain how we will modify the paper.
>
> **Q1: Sec.3.1 and 3.2. are written only for people who are very familiar with the 3 most related papers and has far too many forward and backward pointers.**
>
> We agree that more descriptions of the background should be provided, but due to the page limitation, we have to simplify the introduction of the background methods. We will add more explanations to guide readers' understanding within the space allowed, and will add additional contents with detailed descriptions to the supplementary. As suggested, we will also detail the overall pipeline first and clarify our designs before providing results or comparisons.
>
> **Q2: ‘consistency aware' / ‘field consistency loss' and ‘adversarial optimization' are not explained.**
>
> Indeed, learning the ‘consistency' is a frequently discussed topic in Neural Implicit Functions (e.g. NeRF and SDF). For example, DietNeRF [ICCV 2021] proposed a semantic consistency loss for few-shot view synthesis and SparseNeuS [ECCV 2022] proposed a consistency-aware learning scheme for improving reconstruction quantities. However, due to the different characteristics of implicit fields, there are great differences in learning consistency. In this paper, we focus on learning a consistent UDF with a carefully designed loss. The ‘adversarial optimization’ is having opposite optimization directions which have a highly negative effect on the accuracy and continuity of fields, we will revise our terminology in revision.
>
> **Q3: The real technical contribution. Is the first contribution a very small modification of Neural-Pull?**
>
> Our novelty lies in the analysis of implicit fields which is seldom discussed in previous works. We did get inspiration from Neural-Pull on how to learn distance fields by moving queries. However, the nature of SDF prevents Neural-Pull from representing most real-world objects with open surfaces or geometries with inner structures, and the direct extension of Neural-Pull to UDF fails drastically as shown in Table 5. This observation drives us to design a consistency-aware learning scheme with a carefully designed loss as described in Sec.3.2 which leads to an accurate and continuous field as shown in Fig 1 of the supplementary. In Sec.3.3, we proposed to progressively estimate the mapping relationship between 3D queries and the approximated surface by updating the raw point cloud with well-moved queries as additional priors for promoting further convergence. Finally, previous UDF approaches fail to extract surfaces directly which greatly limits their practicability. We resolve this problem by introducing an algorithm for directly extracting surfaces with arbitrary topology from the gradient vector field of UDF as described in Sec.3.4.
>
> **Q4: The progressive surface approximation seems novel but not claimed clearly.**
>
> We claimed the progressive surface approximation in the introduction (l.48 - l.51) and we will make it more clearly in the revision.
>
> **Q5: The surface extraction seems to be a relatively simple adaptation of marching cube.**
>
> We believe that one of the most important factors preventing the development of UDF approaches is the inability to extract surfaces directly. By observing the gradient vector field of UDF, we propose to classify whether two points are in the same or the opposite side using dot product of the gradients from the learned UDF, and extract surfaces based on this relationship. Our proposed surface extraction algorithm is an efficient method to mesh the UDF, which is not only designed for our method but also suitable for other UDF approaches like NDF [NeurIPS 2020]. As shown in Table 1, the improvement is significant by adopting our surface extraction algorithm to NDF ($NDF_{gradRA}$) than reconstructing the surface from generated dense point clouds using BPA as proposed by NDF ($NDF_{BPA}$).
>
> **Q6: Metrics related to meshes.**
>
> We further report the normal consistency score proposed in OccNet to evaluate the accuracy of meshes on the MGD dataset.
> |methods|Neural-Pull|NDF| Ours|
> |:-:|:-:|:-:|:-:|
> |Normal Consistency|91.83|92.84|**97.80**|
>
> We will add the results in revision.
>
> **Q7: Confusion of the ‘low confidence range' experiment (Table 7).**
>
> The ‘low confidence range’ is the standard deviation of the Gaussian function for sampling auxiliary points. Specifically, as mentioned in l.269 – l.271, a Gaussian function $\mathcal{N}(\mu, \sigma^2)$ with $\mu=p_i$ and $\sigma$ as the distance between $p_i$ and its 50-th nearest points on $P$ is adopt to sample query points for $p_i$ (high confidence range). After the convergence of the first stage, we sample auxiliary points using a Gaussian function with $\sigma^{'} = 1.1\sigma$ (or 0.9, 1.0 and 1.2 as listed in Table 7) for aggressive surface approximation. We will describe the settings and explanations more clearly in revision.

---

> > ### Author Response · Authors · 2022-08-08
> > **Looking Forward to Hearing from Reviewer 1bdf**
> >
> > Dear Reviewer 1bdf,
> >
> > Following your questions, we provided additional explanations and reported the normal consistency score compared to the state-of-the-arts to demonstrate the quality of our reconstructions. We further detailed each of our technical contributions to clarify our ideas and designs, and explained how we will modify the paper as suggested.
> >
> > We would like to discuss with you to further clarify our paper and answer your questions. And if you believe we have addressed your concerns, we hope that you would be willing to increase your score.
> >
> > Thanks for your time,
> >
> > The Authors

---

### Official Review · Reviewer_3mEs · 2022-07-13

**Rating:** 5
**Confidence:** 3
**Soundness:** 3 good
**Presentation:** 3 good
**Contribution:** 3 good

**Summary:**

This paper focuses on the problem of extracting or reconstructing mesh surface from raw point cloud. The key idea is to learn an unsigned distance function to progressively get to the real surface. The unsigned distance field is critical to deal with objects that are not watertight but with inner parts. The authors proposed a consistency aware loss to keep the consistency of the learned unsigned distance fields to avoid adversarial optimization. A surface extraction algorithm is also proposed to extract mesh surface from the learned unsigned distance function.

**Questions:**

I have some questions or confuse on some technical details:
1) Will the optimization fall into local minimum with the Chamfer distance Loss of Eq.(3)? If yes, then how would this local minimum affect the optimization? If no, why the global minimum is guaranteed?
2) what is the performance of directly extending Neural-Pull to unsigned distance field?
3) The authors have spent much efforts on designing the Progressive Surface Approximation, but I didn't see ablation study on this component which I think it is critical.

Also, what is the computational cost to learn the unsigned distance field?

**Limitations:**

The authors have mentioned one limitation or possible future work which is to have a coarse-to-fine divided grids. But they haven't clearly discussed the limitations and also they haven't demonstrated any failure cases. Please refer to the Weakness part about some of my thoughts on the limitations.

**Strengths And Weaknesses:**

Strengths:
First, using unsigned distance function is critical and important to handle complicated object structures. They have demonstrated better performance on public dataset both visually and in numbers.

Weakness:
From my understanding, this proposed method doesn't have potentials to handle any noise in the raw point cloud, which means they require a clean point cloud as input. But in real scenario, the raw point cloud is not noise-free.
Another issue is that the surface extraction algorithm is a bit tricky. The extraction is mainly controlled by computing the sign of cls(.) function, but how could we guarantee the gradient of the unsigned distance field is always accurate.
Finally, the authors haven't presented any failure cases or any dicussion on when the proposed method would fail.

---

> ### Author Response · Authors · 2022-08-02
> **Response to Review 3mEs**
>
> We appreciate the reviewer for his insightful and thorough comments. We will further incorporate the suggestions in the next version.
>
> **Q1: Potentials to handle noises in the raw point clouds.**
>
> We agree that we don’t have special designs for noisy point clouds, especially when learning to reconstruct surfaces from raw point clouds without any supervision. While the related works (e.g. Neural-Pull, NDF, SAL, GIFS) also struggle from handling noisy point clouds, since it is still a challenge to handle clean point clouds. Alternatively, a denoising algorithm (e.g. PointCleanNet) can be used for preprocessing the noisy point clouds first. Furthermore, we have provided the reconstruction results of real scanned shapes and scenes which contain noises with unknown distributions in Sec.4.3 and Sec.4.4, where we significantly outperform the other methods. We also consider this as an interesting future work to reconstruct surfaces from noisy point clouds in an unsupervised way, which will be added in revision.
>
> **Q2: How to guarantee that the gradient is always accurate for surface extraction?**
>
> Indeed, it is extremely difficult to learn a perfect unsigned distance field where the gradient values are guaranteed exactly accurate. However, our proposed surface extraction algorithm only focuses on the direction of gradient which is easy to guarantee since our optimization is conducted by moving queries against the direction of gradient to the approximated surface. Hence, the gradients are highly correlated to the moving direction in the optimization. Eventually, the direction of the gradient can be guaranteed to be broadly correct. Besides, to extract surfaces correctly, we only need to determine whether the gradients at two queries are approximately in the same direction (inner product is positive) or the reverse direction (inner product is negative), which is highly robust.
>
> **Q3: Will the optimization fall into local minimum with the Chamfer distance Loss?**
>
> Our method does not guarantee the global minimum strictly in theory. Actually, since the point cloud is only a discrete representation of the surface, and the topology of the point cloud is ambiguous, it is impossible to converge to an actual global minimum in a strict sense in theory with only raw point clouds as input. What our method guarantees is the consistency of the learned unsigned distance field in contrast to Neural-Pull loss in Eq.2 which will form a distorted field as demonstrated in Fig 3 and Fig 4.
>
> **Q4: What is the performance of directly extending Neural-Pull to unsigned distance field?**
>
> The quantitative results obtained by directly extending Neural-Pull to UDF have been shown in ‘NP loss' of Table 5, and the simulation experiment of this extension has been shown in Fig 4. Furthermore, the visualization of the unsigned distance field learned by Neural-Pull and our method has been shown in Fig 1 in the supplementary. Note that all the designs and experimental settings are kept the same as ours except for the loss. Besides, the quantity and visualization comparisons with the original Neural-Pull which learns SDF were given in Table 2, Table 4, Fig 8 and Fig 9, respectively.
>
> **Q5: The ablation study on the design of Progressive Surface Approximation.**
>
> In Table 6 and Table 7, we have provided the ablation studies for the design of Progressive Surface Approximation. Specifically, we explore the effect of step numbers in Progressive Surface Approximation in Table 6, where we reported the performance of training our network with different numbers of steps St=[1,2,3,4]. And we further explore the range of low confidence regions (described in l.206 – l.210 and l.218 – l.220) about Progressive Surface Approximation in Table 7, where we reported the performance of 4 different range values.
>
> **Q6: What is the computational cost to learn the unsigned distance field?**
>
> We make a comparison with Neural-Pull, IGR, Point2mesh on the computational cost of optimizing for a single point cloud in the following table.
>
> |methods|Neural-Pull|IGR|Point2mesh|Ours|
> |:-:|:-:|:-:|:-:|:-:|
> |Time (s)|1150|1212|4028|**667**|
> |Memory (GB)|2.2|6.1|5.2|**2.0**|
>
> The optimization time is evaluated on a single GTX 3090 GPU. It shows that our method converges faster than all the baselines. We will include the table in the supplementary. We also provided the efficiency comparison of surface generation in Table 2 of the supplementary.
>
> **Q7: Failure cases and discussion on when the proposed method would fail.**
>
> Indeed, our visualization examples are selected randomly from the dataset without checking all the results yet. We will look through all the results later and put the failure cases in revision. We also admit that our method may fail to reconstruct a perfect surface when raw point clouds are extremely noisy or sparse since we don’t require any prior or ground truth supervision, and we will add more discussions of our limitations in the conclusion.

---

> > ### Author Response · Authors · 2022-08-08
> > **Looking Forward to Hearing from Reviewer 3mEs**
> >
> > Dear Reviewer 3mEs,
> >
> > Following your questions, we provided additional explanations and reported the computational cost compared to the state-of-the-arts to illustrate the effectiveness and efficiency of our approach. We further demonstrate our potentials to handle noises by the provided results on the real scanned shapes and scenes.
> >
> > We would like to discuss with you to further clarify our paper and answer your questions. And if you believe we have addressed your concerns, we hope that you would be willing to increase your score.
> >
> > Thanks for your time,
> >
> > The Authors

---

### Comment · Area_Chair_MUDi · 2022-08-08
**Any thoughts from reviewers?**

Hi Reviewers,

The discussion period is closing soon. Please take a look at the responses from the authors. If you have further questions, please ask them now, since the authors will be unable to respond soon. It's substantially more productive, effective, and reasonable to have a quick back-and-forth with authors now than to raise additional questions or concerns post-discussion period that the authors are unable to address.

Thanks,

AC

---

### Meta-Review · Area_Chair_MUDi · 2022-08-22

**Recommendation:** Accept
**Confidence:** Certain

**Metareview:**

All reviewers were clearly in favor of accepting the paper pre-rebuttal. There was limited discussion post-rebuttal. The AC examined the paper, the reviews, and the authors' response and is inclined to accept the paper. The AC encourages the authors to use their extra page to incorporate their responses to the reviewers into the final version of the paper. In particular, the AC would encourage carefully considering the feedback on presentation from 1bdf.

**Award:**

No

---

### Decision · Program_Chairs · 2022-09-14

Accept